# Characterization of aminoglycoside resistance genes in multidrug-resistant *Klebsiella pneumoniae* collected from tertiary hospitals during the COVID-19 pandemic

Omar B. Ahmed[ID][1]*, Atif H. Asghar[1], Majid Bamaga[1], Fayez S. Bahwerth[ID][2], Mutasim E. Ibrahim[ID][3]

1 Department of Environmental and Health Research, The Custodian of the Two Holy Mosques Institute for Hajj and Umrah Research, Umm Al-Qura University, Makkah, Saudi Arabia, 2 King Faisal Hospital, Ministry of Health, Makkah, Saudi Arabia, 3 Department of Basic Medical Sciences (Microbiology Unit), College of Medicine, University of Bisha, Bisha, Saudi Arabia

* abuaglah1@hotmail.com

## Abstract

Since the peak of the coronavirus disease 2019 (COVID-19) pandemic, concerns around multidrug-resistant (MDR) bacterial pathogens have increased. This study aimed to characterize aminoglycoside resistance genes in MDR *Klebsiella pneumoniae* (*K. pneumoniae*) collected during the COVID-19 pandemic. A total of 220 clinical isolates of gram-negative bacteria were collected from tertiary hospitals in Makkah, Saudi Arabia, between April 2020 and January 2021. The prevalence of *K. pneumoniae* was 40.5%; of the 89 *K. pneumoniae* isolates, MDR patterns were found among 51 (57.3%) strains. The MDR isolates showed elevated resistance rates to aminoglycoside agents, including amikacin (100%), gentamicin (98%), and tobramycin (98%). PCR assays detected one or more aminoglycoside genes in 42 (82.3%) MDR *K. pneumoniae* strains. The *rmtD* gene was the most predominant gene (66.7%; 34/51), followed by *aac(6′)-Ib* and *aph(3′)-Ia* (45.1%; 23/51). The *aac(3)-II* gene was the least frequent gene (7.8%; 4/51) produced by our isolates. The *rmtC* gene was not detected in the studied isolates. Our findings indicated a high risk of MDR bacterial infections through the COVID-19 outbreak. Therefore, there is a need for continuous implementation of effective infection prevention control (IPC) measures to monitor the occurrence of MDR pathogens and the emergence of MDR bacterial infections through the COVID-19 outbreak.

## Introduction

Aminoglycosides are a class of antibiotics used to treat patients with a wide range of bacterial infections. They are effective against *Klebsiella pneumoniae* (*K. pneumoniae*) and other gram-negative bacteria in the absence of resistance mechanism determinants [1]. *K. pneumoniae* is one of the leading causes of nosocomial infections in the hospital setting worldwide [2]. This bacterial pathogen might acquire specific virulence plasmids that contain aminoglycoside resistance genes, which may result in serious infections [3, 4].

**Data Availability Statement:** All relevant data are within the paper.

**Funding:** The author(s) received no specific funding for this work.

**Competing interests:** The authors have declared that no competing interests exist.

The SARS-COV-2 coronavirus disease 2019 (COVID-19) pandemic is far from over. Although infections are now less common, many patients with COVID-19 infection are admitted to hospital wards and intensive care units (ICUs). This condition may be accompanied by secondary bacterial infections that require additional empirical broad-spectrum antibiotic treatment. Bacterial coinfection with COVID-19 infection has been documented in the literature [5]. However, viral infections associated with secondary *K. pneumoniae* infection as a part of nosocomial infections can lead to a high mortality rate due to coexistence with respiratory diseases and treatment failure [6]. Evidence from Europe and the United States has reported on the link between the COVID-19 outbreak and an increase in infections with and/ or acquisition of multidrug-resistant (MDR) bacteria [7].

Infection with MDR *K. pneumoniae* has become a threat to public health due to the limited therapeutic options. Aminoglycosides are considered an effective choice in combination with other antibiotic agents to treat MDR *K. pneumoniae* infection [8]. They work by preventing the production of proteins in bacteria by binding to the amino group site of 16S RNAs found inside subunits of the 30S ribosome. This location may make it harder to be coded by the genes, prevent them from being read, and prevent translocation [9, 10]. The most well-known method of resistance to aminoglycosides in these bacteria is inhibiting the development of enzymes [10]. Enzymes that are responsible for modifying aminoglycosides (AMEs) include O-adenyltransferases (ANT), N-acetyltransferases (AAC), and O-phosphotransferases (APH), and they are encoded through DNA molecules known as plasmids. Other AME-encoding genes include *aac (3)-II*, *aac(6′)-I*, *ant(3″)-I*, *aph(3′)-II*, and *ant(2″)-I*. Other mechanisms of resistance to aminoglycosides found in gram-negative bacteria are reducing uptake or decreasing cell permeability in addition to methylating *16S RNA* in ribosomes. However, the *rmtA* gene may control and improve this response. The second process is a reduction in the amount of medication within cells. *K. pneumoniae* also exhibits independent aminoglycoside resistance (i.e., inactivating enzymes). Due to an efflux mechanism that lowers the amount of aminoglycosides, this resistance is distinguished by engaging all other forms of aminoglycosides [11]. To varying degrees, the enzymes *aac(6′)-I*, *aac(3)-II*, *aph(3′)-II*, *ant(3)-I*, and *ant(2)-I* may enhance resistance to aminoglycosides [10]. Both COVID pandemic and non-COVID respiratory viral infections could affect antibiotic use, by increasing antibiotic sales with the easing of lockdowns which may have contributed to increased antibiotics sales during covid-19 pandemic. The outbreak of MDR *K. pneumoniae* during the COVID-19 pandemic has been reported in many parts of the world [12–14], but very few laboratory-based reports have been published. The present study was carried out to identify the frequency of aminoglycoside resistance genes in clinical isolates of MDR *K. pneumoniae* collected from tertiary hospitals during the COVID-19 pandemic.

## Materials and methods

A descriptive cross-sectional laboratory-based study designed to be conducted at tertiary hospitals (5 hospitals in Makkah) for one-year period. The Yamane formula used for determining the necessary sample size is as follows: $n = N(1+Ne^2)$, where n = sample size (= 220), N = population size (480 based on the total visits of the patients in the 5 tertiary hospitals), and e (significance) = 0.05. A total of 220 isolated pathogenic gram-negative bacilli were collected from the laboratory departments at 5 tertiary hospitals in Makkah, Saudi Arabia, from April 2020 to January 2021.

### Bacterial identification

The 220 isolates were collected from outpatients, inpatients, and ICU patients during the period of COVID-19 pandemic. The isolates were collected from the microbiology laboratory

of the hospitals during routine investigation of the disease; therefore, ethical consent is not required per the nature of the study. Bacterial isolates were identified using a VITEK 2 Compact System using AST-GNI cards (Biomerieux). DensiCHEK Plus was established within a 0.5 McFarland standard in a 0.48% sterile sodium chloride solution. The time interval between suspension preparation and card filling was less than 30 min to avoid changes in turbidity. The bacterial suspension was manually loaded using cards for the VITEK 2 system. The system automatically filled each test card, sealed it, and incubated it for approximately 3 hours. The system then analyzed the data after repeated kinetic fluorescence measurements before automatic result reporting.

## Antibiotic susceptibility testing

The susceptibilities of the *K. pneumoniae* isolates to 11 selected antibiotics of different classes, including aminoglycosides, were tested using the VITEK 2 system (AST no. 12 card). The disc-diffusion method was also performed in accordance with the guidelines of the Clinical and Laboratory Standards Institute (CLSI) [12]. The following antibiotics were examined: amikacin (AK; 30 μg), gentamicin (GN; 10 μg), tobramycin (TOB; 10 μg), amoxicillin clavulanic acid (AMC; 20/10 μg), ciprofloxacin (CIP; 5 μg), cefepime (CPM; 30 μg), imipenem (IMP; 10 μg), cefotaxime (CTX; 30 μg), ampicillin (AMP; 10 μg), aztreonam (ATM; 30 μg), and cefuroxime (CXM; 30 μg). The isolates were classified as susceptible, intermediate, or resistant to each antibiotic according to CLSI guidelines [15]. MDR patterns in *K. pneumoniae* were defined as acquired nonsusceptibility to at least one agent in three or more antibiotic classes [16]. *K. pneumoniae* isolates showing an MDR pattern and resistance to one of three aminoglycoside agents (amikacin, gentamicin, or tobramycin) were subsequently submitted for PCR amplification.

## PCR analysis

Multiplex PCR assays were performed to detect the following nine aminoglycoside resistance genes: *aac(6′)*-Ib, *aac(3)-II*, *aph(3′)-Ia*, *rmtA (3)*, *rmtB*, *rmtC*, *armA*, *rmtD*, and *npmA*. In brief, an aliquot of cell suspension containing $10^7$ cells/mL from each MDR *K. pneumoniae* isolate was used for DNA extraction by the boiling method as previously described [17]. The collected material was placed into a tube containing 50 μL nuclease-free water and boiled at 100˚C for 10 minutes. The mixture was cooled in ice and then centrifuged at $3000 \times$ g for 10 seconds. The upper layer containing DNA was used for PCR [17]. The isolates were then subjected to PCR analysis to detect the target resistance genes. PCR was performed in a final volume of 25 μL, containing 10 μL of PCR master mix (Promega Corporation, Madison, USA), 1.5 μL of each primer, 4 μL of template DNA, and 9.5 μL of purified distilled water. The primers used for PCR analysis are listed in Table 1. Two multiplex reactions were performed. The PCR conditions were as follows: predenaturation at 94˚C for 4 minutes; 30 amplification cycles of 94˚C for 30 seconds (denaturation), annealing temperature (shown in Table 2) for 30 seconds, and 72˚C for 1 minute (extension); and 72˚C for 10 minutes (final extension). PCR products were analyzed with 1% agarose gel electrophoresis. A DNA ladder (100 bp) was used with each run, and the gene was determined by the size of the amplified PCR product.

## Statistical analysis

Statistical analysis was performed using SPSS software (v.25, IBM, United States). Simple descriptive statistics are presented in the form of numbers, proportions, and frequencies.

**Table 1. Primers used for PCR detection of resistance genes among MDR *Klebsiella pneumoniae*.**

| No. | Target gene | Primer | Sequence (5′–3′) | Annealing temperature | Product size (bp) | Ref. |
|---|---|---|---|---|---|---|
| 1. | *aac(6′)-Ib* | aac(6′)-Ib-F | TG CGA TGC TCT ATG AGT GGC TA | 56 | 472 | [18] |
| | | aac(6′)-Ib-R | CTC GAA TGC CTG GCG TGT TT | | | |
| 2. | *aac(3)-II* | aac(3)-II-F | ATATCGCGATGCATACGCGG | 56 | 877 | [19] |
| | | aac(3)-II-R | GACGGCCTCTAACCGGAAGG | | | |
| 3. | *aph(3′)-Ia* | aph(3′)-Ia-F | CGAGCATCAAATGAAACTGC | 50 | 623 | [20] |
| | | aph(3′)-Ia-R | GCGTTGCCAATGATGTTACAG | | | |
| 4. | *rmtA (3)* | rmtA-F | CTA GCG TCC ATC CTT TCC TC | 58 | 635 | [21] |
| | | rmtA-R | TTT GCT TCC ATG CCC TTG CC | | | |
| 5. | *rmtB* | rmtB-F | GCT TTC TGC GGG CGA TGT AA | 55 | 173 | [22] |
| | | rmtB-R | ATG CAA TGC CGC GCT CGT AT | | | |
| 6. | *rmtC* | rmtC-F | CGA AGA AGT AAC AGC CAA AG | 55 | 711 | [22] |
| | | rmtC-R | ATC CCA ACA TCT CTC CCA CT | | | |
| 7. | *armA* | armA-F | ATT CTG CCT ATC CTA ATT GG | 55 | 315 | [22] |
| | | armA-R | ACC TAT ACT TTA TCG TCG TC | | | |
| 8. | *rmtD* | rmtD-F | CGG CAC GCG ATT GGG AAG C | 40 | 401 | [22] |
| | | rmtD-R | CGG AAA CGA TGC GAC GAT | | | |
| 9. | *npmA* | npmA-F | CTC AAA GGA ACA AAG ACG G | 58 | 774 | [23] |
| | | npmA-R | GAAACATGGCCAGAAACTC | | | |

## Results

Gram-negative bacteria were obtained from patients' clinical samples of sputum (n = 81), urine (n = 57), wound (n = 45), blood (n = 32), and cerebrospinal fluid (n = 5). Most of the isolates were *K. pneumoniae* (89; 40.5%), followed by *Escherichia coli* (58; 26.4%) and *Pseudomonas aeruginosa* (23; 10.5%) (Table 2). Of the 89 *K. pneumoniae* isolates, MDR patterns were found in 51 (57.3%) strains.

The susceptibility testing results for the MDR *K. pneumoniae* (n = 51) isolates against different antibiotic classes are shown in Table 3. The MDR isolates showed a high resistance rate to the aminoglycoside agents amikacin (100%), gentamicin (98%), and tobramycin (98%). MDR isolates (51) expressed total resistance (100%) for amoxicillin clavulanic acid, ciprofloxacin, cefotaxime, ampicillin, aztreonam, and cefuroxime, while the resistance rate for cefepime was 98%. The MDR isolates showed the lowest resistance rate for imipenem (74.5%) (Table 3).

Of the 51 MDR *K. pneumoniae* isolates, a multiplex PCR assay detected one or more aminoglycoside genes in 42 (82.3%) isolates (Fig 1). A single gene was detected in 7 (13.8%) isolates,

**Table 2. Frequency of gram-negative bacteria collected from clinical samples of the patients.**

| Isolate | n (%) |
|---|---|
| *K. pneumoniae* | 89 (40.5%) |
| *Escherichia coli* | 58 (26.4%) |
| *Pseudomonas aeruginosa* | 23 (10.5%) |
| *Acinetobacter baumannii* | 19 (8.6%) |
| *Proteus mirabilis* | 9 (4.1%) |
| *Klebsiella oxytoca* | 8 (3.6%) |
| *Enterobacter* spp. | 8 (3.6%) |
| *Chryseobacterium indologenes* | 6 (2.7%) |
| Total | 220 (100%) |

**Table 3. Resistance rates of MDR *K. pneumoniae* isolates tested against different antibiotic classes.**

| Class | Agent | Susceptible | Intermediate | Resistant |
|---|---|---|---|---|
| Aminoglycosides | Amikacin | 0 (0%) | 0 (0%) | 51(100%) |
| | Gentamicin | 1 (2%) | 0 (0%) | 50 (98%) |
| | Tobramycin | 0 (0%) | 1 (2%) | 50 (98%) |
| Cephalosporins | Cefepime | 0 (0%) | 2 (4%) | 49 (96%) |
| | Cefotaxime | 0 (0%) | 0 (0%) | 51 (100%) |
| | Cefuroxime | 0 (0%) | 0 (0%) | 51(100%) |
| Carbapenems | Imipenem | 11 (21.5%) | 2 (4%) | 38 (74.5%) |
| Fluoroquinolones | Ciprofloxacin | 0 (0%) | 0 (0%) | 51 (100%) |
| Monobactams | Aztreonam | 0 (0%) | 0 (0%) | 51 (100%) |
| Penicillins | Ampicillin | 0 (0%) | 0 (0%) | 51 (100%) |
| Penicillins + b-lactamase inhibitors | Amoxicillin clavulanic acid | 0 (0%) | 0 (0%) | 51 (100%) |

whereas 35 (68.6%) isolates harbored two or more aminoglycoside genes. Fig 2 illustrates the proportions of aminoglycoside genes among the MDR *K. pneumoniae* isolates (n = 42). The *rmtD* gene was the most common gene detected among MDR *K. pneumoniae* isolates (66.7%; 34/51), followed by *aac(6′)-Ib* and *aph(3′)-Ia* (45.1%; 23/51). The *aac(3)-II* gene was the least frequent gene (7.8%; 4/51) produced by our isolates. No *rmtC* gene was detected in the studied isolates.

## Discussion

The COVID-19 pandemic may lead to the emergence of other infections as well as MDR bacterial pathogens. The dissemination of pathogenic *K. pneumoniae* resistant to aminoglycosides in hospital settings may lead to difficulties in treatment and increase the mortality rate. We aimed to identify the prevalence of genes encoding resistance to aminoglycosides in *K. pneumoniae* clinical isolates during the COVID-19 pandemic. In the present study, all MDR *K. pneumoniae* isolates showed complete resistance to amikacin (100%) and a high resistance rate to other tested aminoglycoside antibiotics. Worldwide, the aminoglycoside resistance rate of *K. pneumoniae* has been increasing due to the overuse of aminoglycosides and since then, it

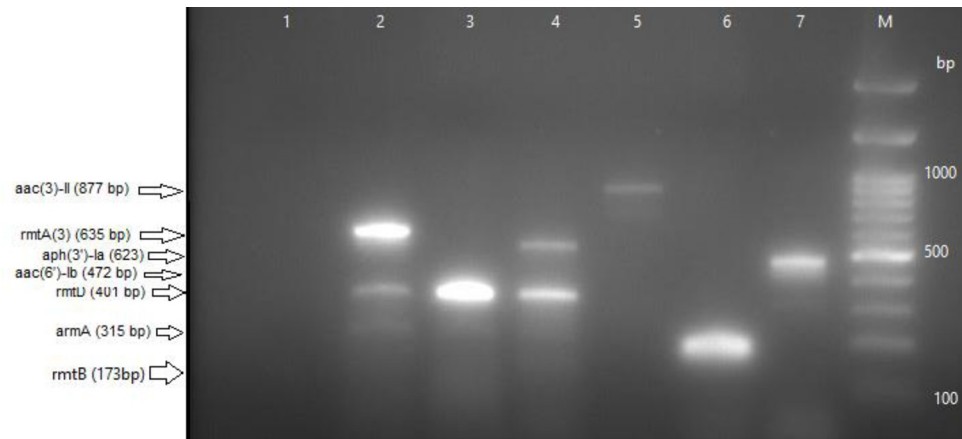

**Fig 1. Detection of aminoglycoside genes on 1.2% agarose gel electrophoresis.** Lane 1 showed a negative result; Lane 2 was positive for three genes (*rmtA (3)-Ia*, *rmtD*, *armA*); Lane 3 was positive for *rmtD*; Lane 4 was positive for two genes (*aph(3′)-Ia* and *rmtD*); Lane 5 was positive for *aac(3)-II*; Lane 6 was positive for *rmtB*; and Lane 7 was positive for *aac(6′)-Ib*. Lane M was a 100 bp DNA ladder.

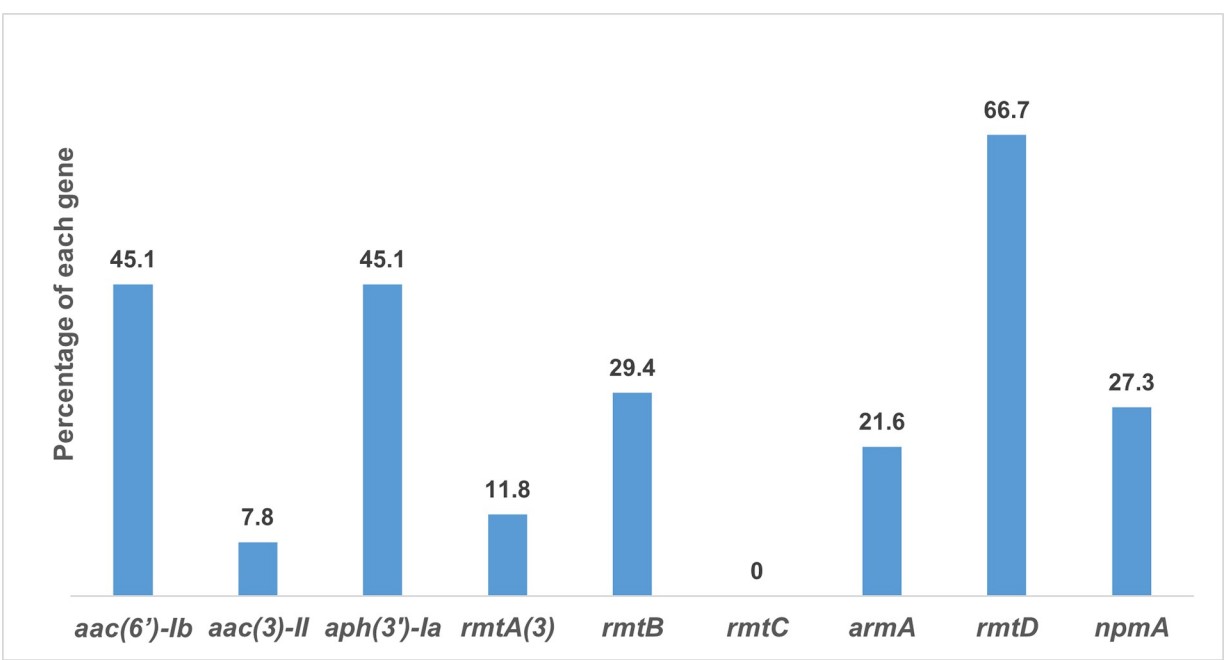

**Fig 2. Distribution of aminoglycoside resistance genes among MDR *K. pneumoniae* isolates.**

has broken out in many countries [24, 25]. Amikacin, gentamycin and tobramycin are typical aminoglycosides for treating infections caused by gram-negative bacteria in Saudi Arabia [21]. Amikacin was shown to be the most efficient antibiotic against common gram-negative bacteria in Makkah, despite high levels of antibiotic resistance reported by others [26]. The emergence of aminoglycoside-resistant *K. pneumoniae* strains during the COVID-19 pandemic has been described in many studies, and MDR *K. pneumoniae* may be associated with both COVID-19 ICU and non-COVID-19 ICU patients [12–14]. The frequent use of antibiotics and extended ICU stays were identified as common risk factors for outbreak situations. Moreover, the excessive use and misuse of antibiotics for viral respiratory infections may be the reason for elevated resistance rates [7]. Arteaga-Livias et al. [27] suggested that cross-contamination via the hands of the staff and the inappropriate use of personal protective equipment might facilitate the spread of MDR bacteria amid the COVID-19 pandemic. Our results revealed eight diverse PCR patterns in 42 (82.3%) of the MDR *K. pneumoniae* isolates. The coproduction of more than one aminoglycoside gene was detected in 68.6%, while single genes were detected in 13.7% of the isolates. The widespread distribution of plasmids in *K. pneumoniae*, which serve as sources of resistance acquisition through lateral gene transfer, may be related to resistance against aminoglycosides. However, any modifications made by genes encoding aminoglycoside-modifying enzymes (AMEs) may decrease the binding affinity of drugs for their targets and hence lead to a loss in antibacterial effects [27].

In the present study, the most predominant aminoglycoside gene was *rmtD*, followed by *aac(6′)-Ib* and *aph(3′)-Ia*. These findings contradict a previous study in Makkah, which reported that *rmtB* and *aac(6′)-Ib* were the most common genes [28]. Globally, before the COVID-19 pandemic, the prevalence of aminoglycoside genes was variable. A study in Spain reported that *aac(6′)-Ib* was the most commonly detected aminoglycoside gene, followed by *aph(3′)-Ia* and *aac(3)-IIa*, in *Escherichia coli* and *K. pneumoniae*, producing extended-spectrum b-lactamases [29]. In Australia, the most frequent genes were *aac(3)-IIa* and *aac(6′)-Ib*

[30]. In Norway, the most prevalent AME gene was *aac(6′)-Ib*, followed by *aac(3)-IIa* [29]. In China, the most prevalent AME gene in *K. pneumoniae* was *aac(3)-II*, followed by *aac(6′)-Ib*, *armA*, and *rmtB* [30]. Multiple aminoglycoside resistance genes were reported in Brazil and Mexico [31, 32]. In addition, several recent studies have also reported an increase in MDR during the COVID-19 pandemic [33–35].

A major concern of this study was the presence of various aminoglycoside-resistance gene types, singly and in combination, in isolates, which is consistent with other findings [36]. This may be explained by the incidence and persistence of resistance genes within *K. pneumoniae*, which may be facilitated by antibiotic stress in medical settings, particularly during large gatherings (such as pilgrimages) [37]. Although PCR method may has some limitations such as false-positive results due to DNA contamination; detection sensitivity may exceed clinical significance; and limited detection space of pathogen identification, but it could be an essential tool for rapid and sensitive amplification of resistance genes (genotypic resistance) and even studying of multiple microbiomes. In future studies, in-depth and accurate data from whole genome sequencing can helps detect and track epidemics earlier and with the PCR, it can be also utilized in identifying resistance genes, track outbreaks, and even enhancing surveillance methods in bacteria.

## Conclusions

The study concluded that MDR *K. pneumoniae* isolates collected during the COVID-19 pandemic in Makkah hospitals were highly resistant to aminoglycosides and other commonly used antibiotics. Diverse aminoglycoside resistance genes, singly or in combination, were detected in MDR isolates. *rmtD* was the most prevalent gene, followed by the *aac(6′)-Ib* and *aph(3′)-Ia* genes. Our findings indicated that there is a substantial risk of emerging MDR bacterial infections through the COVID-19 outbreak. Therefore, the implementation of effective infection prevention control measures should be continuous to monitor the occurrence of MDR pathogens in such situations.

## Author Contributions

**Conceptualization:** Omar B. Ahmed, Atif H. Asghar.

**Data curation:** Majid Bamaga, Fayez S. Bahwerth.

**Formal analysis:** Omar B. Ahmed, Atif H. Asghar.

**Investigation:** Mutasim E. Ibrahim.

**Methodology:** Omar B. Ahmed, Fayez S. Bahwerth, Mutasim E. Ibrahim.

**Project administration:** Majid Bamaga.

**Resources:** Omar B. Ahmed, Majid Bamaga, Fayez S. Bahwerth.

**Software:** Majid Bamaga, Fayez S. Bahwerth.

**Supervision:** Atif H. Asghar.

**Validation:** Omar B. Ahmed, Majid Bamaga, Mutasim E. Ibrahim.

**Visualization:** Atif H. Asghar.

**Writing – original draft:** Omar B. Ahmed, Fayez S. Bahwerth, Mutasim E. Ibrahim.

**Writing – review & editing:** Omar B. Ahmed, Atif H. Asghar, Mutasim E. Ibrahim.

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
