## [Decision Letter · Decision Letter 0]

20 Feb 2023

PONE-D-23-00999Characterization of aminoglycoside-resistant genes in multidrug-resistant Klebsiella pneumoniae collected from tertiary hospitals during the COVID-19 pandemicPLOS ONE

Dear Dr. Ahmed,

Thank you for submitting your manuscript to PLOS ONE. After careful consideration, we feel that it has merit but does not fully meet PLOS ONE’s publication criteria as it currently stands. Therefore, we invite you to submit a revised version of the manuscript that addresses the points raised during the review process.

We look forward to receiving your revised manuscript.

Kind regards,

Mabel Kamweli Aworh, DVM, MPH, PhD. FCVSN

Academic Editor

PLOS ONE

Journal Requirements:

" ext-link-type="uri" xlink:type="simple">https://journals.plos.org/plosone/s/file?id=ba62/PLOSOne_formatting_sample_title_authors_affiliations.pdf"

https://covid19-data.nist.gov/pid/rest/local/paper/increasing_frequency_of_aminoglycoside_resistant_klebsiella_pneumoniae_during_the

https://pubmed.ncbi.nlm.nih.gov/34075332/

In your revision ensure you cite all your sources (including your own works), and quote or rephrase any duplicated text outside the methods section. Further consideration is dependent on these concerns being addressed.

Additional Editor Comments:

1. In the discussion section, the authors should kindly provide the interpretation of their study results while comparing these to other published works. Please do not repeat the results in the discussion section as these have already been reported in the results section rather provide a possible explanation for your findings.

2. The authors need to highlight the limitations of their study.

3. There are two concluding sections in this manuscript. The authors need to merge these into one conclusion section, please.

Reviewers' comments:

Reviewer's Responses to Questions

**Comments to the Author**

1. Is the manuscript technically sound, and do the data support the conclusions?

Reviewer #1: Partly

Reviewer #2: Yes

Reviewer #3: Partly

2. Has the statistical analysis been performed appropriately and rigorously? 

Reviewer #1: No

Reviewer #2: Yes

Reviewer #3: Yes

3. Have the authors made all data underlying the findings in their manuscript fully available?

Reviewer #1: Yes

Reviewer #2: Yes

Reviewer #3: No

4. Is the manuscript presented in an intelligible fashion and written in standard English?

Reviewer #1: No

Reviewer #2: No

Reviewer #3: No

5. Review Comments to the Author

Reviewer #1: The study is an interesting one and if more work is put in, it would be great for publishing. There are concerns with the title with regards the work done, findings, discussion, and conclusion hence can’t be published in its current state.

Here are some comments to help improve the quality of the manuscript.

1. Line 2, page 8; please use coronavirus disease (COVID-19) pandemic instead of coronavirus disease -19 (COVID-19)

2. Line 3, page 8; the aim of the study does not correspond to the title

3. Line 6, page 8; please remove “overall”

4. Line 7, page 8; please correct “Of” to “of”

5. Line 12, page 7; please use the word “respectively” and not “each”

6. Line 12, page 8; please change “less frequent” to “least frequent”

7. Line 11, page 8; is there any reason that sentence is italicized? If none, please change to regular letters

8. Line 11, page 8; please maintain consistency in the use of denominators throughout the

manuscript it would help in clarifications

9. Line 13, page 8; please review this statement. As stated in your findings, is the increase as a result of increased reporting followed a none or decreased reporting era or an actual increase in numbers during this period?

10. Line 14, page 8; please review the use of the word “emerging”

11. Line 15, page 8; please change statement to “for continuous implementation of effective infection prevention control (IPC) measures…..”

12. Line 17, page 8; please revise and or review the inclusion of COVID-19 pandemic as a keyword. The study does not show strong attribution to coronavirus disease (SARS-COV-2) unless if the 220 samples were from ONLY COVID-19 positive patients or if the use was just to describe an era? This was not stated in the description of your samples

13. Line 25, page 8; please remove 2019 and leave as coronavirus disease (COVID-19)

14. Line 28, page 8; please include SARS-COV-2 in bracket after the full name

15. Line 32, page 8; please change “Europe countries” to “Europe”

16. Line 44, page 9; please correct “others” to “other”

17. Line 45, page 9; please remove “also” so it reads “Other mechanisms”

18. Line 56, page 9; please remove “aimed to identify” it is a tautology

19. Lines 61-64, page 9; there is no proper description as to the nature of the isolates for example were they from only COVID-19 positive cases, were they outpatients or inpatients or from ICU patients hence was it a healthcare associated infection or community acquired these would shed more light to the correlation

20. Line 64, page 9; please what disease were you referring to in this sentence for clarity

21. Line 68, page 10; please use acceptable short forms for units. 30 min should be 30 mins or 30 minutes

22. Line 70, page 10; please use acceptable short forms for units 3 h should be 3 hrs. or 3 hours

23. Lines 81-85, page 10; please rewrite and give clarity to the statement

24. Line 93, page 10; please use acceptable short forms for units. 10 s should be 10 secs or 10 seconds. please be uniform throughout your writing this includes line 99 and any others

25. Line 93, page 10; please correct “the isolates then were” to “the isolates were then”

26. Line 108, page 11; 220 should be in figures not words, write numbers in figures not words

27. Line 115, page 11; the use of the phrase “elevated resistance rates” should be reviewed since there is no comparism (with a previous result or another study). The phrase “high resistance rate” is more appropriate in this case.

28. Line 117, page 11; please use the word “total” in place of “complete”

29. Line 121-125, page 11; the name K. pneumoniae should be in italics

30. Line 126, page 11; please replace the word “less” with “least”

31. The discussion section should be rewritten please, with clarity on aim of the study and the title and the importance of COVID-19 in the study. As stated in the discussion, is the increase as a result of increased reporting (in this case due to active case search) or an actual increase in numbers of cases with time?

32. It may also help to review the references and use more recent ones with SARS-COV-2 involved directly to bring out a strong attribution. Of the 33 references used, only 8 were from the COVID pandemic era (2020-2021) and only 4 of the 8 had COVID-19 positive cases this is important based on your title. The results should not be repeated but interpreted and compared to other studies to further strengthen your aim. Some sentences are incomplete or may be contradictory e.g., “whereas MDR K. pneumoniae may be associated with both COVID-19 ICU and non-COVID-19 ICU patients [12] [13] [14]. The frequent use of antibiotics and extended ICU stays were identified as common risk factors for outbreak situations. Moreover, the excessive use and misuse of antibiotics for viral respiratory infection may be the reason for elevated resistance rates” please shed more light

33. Page 16, Conclusion; please correct the statement “there are need in continuous implementation of effective control measures” to “Therefore implementation of effective infection prevention control measures should be continuous in order to monitor the occurrence of MDR pathogens in such situations”, or use “there is need.’. Is this statement suggestive that IPC measures be restricted to COVID-19? Kindly provide some recommendations based on your findings also the data provided should support the title of the study

34. If this research is to be published, the title should be reviewed; the attribution to COVID-19 is weak. Though samples were collected during the COVID-19 pandemic nothing strong suggesting resistance occurred as a result of coronavirus infection.

35. The work could be presented without the unnecessary attribution to COVID-19 except the isolates were strictly from ONLY COVID-19 positive cases which wasn’t stated.

Reviewer #2: The authors of the study explored the frequency and categories of aminoglycoside resistance genes in multi-drug resistant (MDR) Klebsiella pneumonia samples collected from 5 tertiary hospitals in Makkah, Saudi Arabia during the COVID pandemic. MDR K. Pneumonia is challenging to treat, and a leading cause of hospital-acquired infections. In patients with co-occurrence of MDR K. Pneumonia and COVID-19 or other respiratory infections, this may heighten the risk for mortality.

The isolates analyzed were human biology specimens (urine, saliva, wound, etc.) that were collected as part of routine treatment care of the patients and not primarily for research. A total of 220 clinical isolates of gram-negative bacteria were collected from April 2020 to January 2021. The authors found out that 89 (40.5%) of these isolates had K. pneumonia, out of which 51 (57.3%) had patterns of MDR. All the MDR K. pneumonia isolates showed resistance to aminoglycoside agents Amikacin (100%), Gentamycin (98%), and Tobramycin (98%). Further PCR analysis showed that 42 isolates out of these 51 had one or more aminoglycoside resistance genes, with some target resistant genes more predominant than others. For instance, the rmtD gene was the most frequent versus the rmtC gene which was not found in any of the isolates. The authors conclude by proposing increased infection control surveillance to monitor the occurrence of MDR K. pneumonia.

The major strength of this study is that it is one of the few published studies that measure the frequency and characterizes aminoglycoside-resistant genes in K.Pneumonia during the COVID era and as such it contributes new insight. Others include; the title and abstract being appropriate for the context of the article; the use of PCR analysis to complement the Viteck-2 Compact System in the study methodology; the authors also did a good job of relating their study findings to other previous studies.

Areas for improvement

The authors should read the article carefully to correct all grammatical errors and typos e.g. Line 3 under PCR Analysis has "descried" instead of described, The last sentence under Introduction has “ identify aimed to identify the”… and many others.

There appears to be a mix-up of tables 1 and 2 in the description under the results section.

The authors should check the frequencies and percentages of one or more aminoglycoside genes and the single gene detected in the MDR K pneumonia – 35/42 and 7/42 as opposed to 35/51 and 7/51.

There were two concluding sections.

No limitations of the study were identified for example PCR has its limitations - specificity, sensitivity, etc. and there are newer methods such as whole-genome sequencing, etc.

Reviewer #3: ABSTRACT

Line 1-2: “The occurrence of multidrug resistant (MDR) bacterial pathogens may become a significant worry, particularly during the last coronavirus disease-19 (COVID-19) pandemic.”

Comment: This sentence needs to be re-constructed. The pandemic has not ended. It has only slowed. Here is a suggestion: "Since the peak of the coronavirus disease-19 (COVID-19) pandemic, concerns around multidrug resistant (MDR) bacterial pathogens have increased"

Line 4-6: “This bacterial pathogen might acquire specific virulence plasmids that contain aminoglycoside-resistant genes, which may result in the dissemination of serious infections”

Comment: dissemination is not the most appropriate word in this sentence

Line 14-16: “Therefore, there is a need for continuous implementation of effective control measures to monitor the occurrence of MDR pathogens in such situations.”

Comment: In which situation? Since it is abstract, it is helpful to be explicit.

INTRODUCTION

Paragraph 1- Line 7: “During the last coronavirus disease 2019 (COVID-19) pandemic..”

Comment: Again, the pandemic is far from over. Although infections are now far between

Paragraph 1- Line 13-15: “Evidences from Europe countries and the United States have reported on the link between the COVID-19..”

Comment: European countries and not Europe countries

Paragraph 2- Line 13: “Other also mechanisms of resistance to aminoglycosides found in Gram-negative bacteria are uptake reduction or decreasing cell permeability besides methylating 16S RNA in ribosomes.”

Comment: Delete “Other” from the beginning of this sentence

Paragraph 2- Line 21-24: “The present study was carried out to identify aimed to identify the frequency of aminoglycoside resistance genes in clinical isolates of MDR K. pneumoniae collected from tertiary hospitals during the COVID19 pandemic”

Comment: Delete “identify aimed to”

Comment: Break the large paragraphs in the introduction to make it easier to read.

MATERIALS AND METHODS

Bacterial Identification

Comment: There was no justification for why the samples were collected between April 2020 and January 2021. Why the choice of hospital too? This was not stated. Why did the research stop at collecting 220 pathogenic Gram-negative bacilli? Why not more or less? The result section provided adequate visuals to convey the insights from the data collection and analysis procedure.

Comment: there are two conclusion sections and that is not helpful. One conclusion is enough for the article.

“second” CONCLUSION section

Paragraph 1: Line 1: “The study concluded that MDR K. pneumoniae collected during COVID-19 pandemic in Makkah hospitals was highly resistant to aminoglycosides and other commonly used antibiotics.”

Comment: “...during COVID_19…” should be rewritten to reflect that the period of data collection was at the peak of the pandemic and COVID-19 is far from over. The way it is phrased here indicates a past occurrence.

General comments

The objective of the study was clearly stated

The method section was underwhelming. The authors did not provide enough information about their process?

The researchers mentioned that the project had no ethical implication, but the pathogenic Gram-negative bacilli were collected from human fluids and their gender split was recorded. How did this scale through the institutional review board of their affiliated institutions of higher learning?

The first paragraph of the result section should be included in the materials and methods section.

The conclusion section was hurried.

The manuscript should be proofread and rearranged for cohesive flow.

6. PLOS authors have the option to publish the peer review history of their article (what does this mean?). If published, this will include your full peer review and any attached files.

Reviewer #1: No

Reviewer #2: No

Reviewer #3: No

---

## [Author Response · Author response to Decision Letter 0]

14 Apr 2023

The authors have responded to all reviewers comments (see attached document).

---

## [Decision Letter · Decision Letter 1]

7 Jun 2023

PONE-D-23-00999R1Characterization of aminoglycoside resistance genes in multidrug-resistant Klebsiella pneumoniae collected from tertiary hospitals during the COVID-19 pandemicPLOS ONE

Dear Dr. Ahmed‬‏,

Thank you for submitting your manuscript to PLOS ONE. After careful consideration, we feel that it has merit but does not fully meet PLOS ONE’s publication criteria as it currently stands. Therefore, we invite you to submit a revised version of the manuscript that addresses the points raised during the review process.

We look forward to receiving your revised manuscript.

Kind regards,

Mabel Kamweli Aworh, DVM, MPH, PhD. FCVSN

Academic Editor

PLOS ONE

Journal Requirements:

Reviewers' comments:

Reviewer's Responses to Questions

**Comments to the Author**

1. If the authors have adequately addressed your comments raised in a previous round of review and you feel that this manuscript is now acceptable for publication, you may indicate that here to bypass the “Comments to the Author” section, enter your conflict of interest statement in the “Confidential to Editor” section, and submit your "Accept" recommendation.

Reviewer #1: All comments have been addressed

Reviewer #2: (No Response)

2. Is the manuscript technically sound, and do the data support the conclusions?

Reviewer #1: Yes

Reviewer #2: Yes

3. Has the statistical analysis been performed appropriately and rigorously? 

Reviewer #1: Yes

Reviewer #2: Yes

4. Have the authors made all data underlying the findings in their manuscript fully available?

Reviewer #1: Yes

Reviewer #2: Yes

5. Is the manuscript presented in an intelligible fashion and written in standard English?

Reviewer #1: Yes

Reviewer #2: Yes

6. Review Comments to the Author

Reviewer #1: The author has made adequate corrections and addressed my points appropriately hence manuscript is good enough for submission and acceptance

Reviewer #2: Thank you for addressing the previous comments. The article is now clearer, however, some revisions still need to be made.

Lines 182-184: this statement is not clear and the two references refer to a pre-COVID era. The authors should clarify the statement.

Lines 219-221: The flow would be better if this section was moved somewhere after Conclusions.

Lines 221-227: Although the authors attempted to address an earlier comment, there is room for further improvement in this section. The current presentation of the limitations of PCR and the advantages of whole genome sequencing seems to undervalue the use of PCR in the analysis, potentially devaluing the entire study. It would be beneficial to provide a more balanced discussion that acknowledges the limitations of PCR but also justifies its usage in the context of this study in conjunction with highlighting the advantages of whole genome sequencing for future studies.

Line 221:the authors should correct the typo in "methjod"

7. PLOS authors have the option to publish the peer review history of their article (what does this mean?). If published, this will include your full peer review and any attached files.

Reviewer #1: **Yes: **Folashade Onatola Toye

Reviewer #2: No

---

## [Author Response · Author response to Decision Letter 1]

9 Jun 2023

authors have responded to reviewers (see attached)

---

## [Decision Letter · Decision Letter 2]

18 Jul 2023

Characterization of aminoglycoside resistance genes in multidrug-resistant Klebsiella pneumoniae collected from tertiary hospitals during the COVID-19 pandemic

PONE-D-23-00999R2

Dear Dr. AHMED,

We’re pleased to inform you that your manuscript has been judged scientifically suitable for publication and will be formally accepted for publication once it meets all outstanding technical requirements.

Kind regards,

Mabel Kamweli Aworh, DVM, MPH, PhD. FCVSN

Academic Editor

PLOS ONE

Additional Editor Comments (optional):

Reviewers' comments:

Reviewer's Responses to Questions

**Comments to the Author**

1. If the authors have adequately addressed your comments raised in a previous round of review and you feel that this manuscript is now acceptable for publication, you may indicate that here to bypass the “Comments to the Author” section, enter your conflict of interest statement in the “Confidential to Editor” section, and submit your "Accept" recommendation.

Reviewer #1: All comments have been addressed

Reviewer #2: All comments have been addressed

2. Is the manuscript technically sound, and do the data support the conclusions?

Reviewer #1: Yes

Reviewer #2: (No Response)

3. Has the statistical analysis been performed appropriately and rigorously? 

Reviewer #1: Yes

Reviewer #2: (No Response)

4. Have the authors made all data underlying the findings in their manuscript fully available?

Reviewer #1: No

Reviewer #2: (No Response)

5. Is the manuscript presented in an intelligible fashion and written in standard English?

Reviewer #1: Yes

Reviewer #2: (No Response)

6. Review Comments to the Author

Reviewer #1: The authors have made a great effort in making corrections to the manuscript . All my concerns have been addressed appropriately

Reviewer #2: (No Response)

7. PLOS authors have the option to publish the peer review history of their article (what does this mean?). If published, this will include your full peer review and any attached files.

Reviewer #1: **Yes: **Folashade Onatola Bamidele

Reviewer #2: No

---

## [Editor Report · Acceptance letter]

19 Jul 2023

PONE-D-23-00999R2 

Characterization of aminoglycoside resistance genes in multidrug-resistant *Klebsiella pneumoniae* collected from tertiary hospitals during the COVID-19 pandemic 

Dear Dr. Ahmed‬‏:

I'm pleased to inform you that your manuscript has been deemed suitable for publication in PLOS ONE. Congratulations! Your manuscript is now with our production department. 

Kind regards, 

on behalf of

Dr. Mabel Kamweli Aworh 

Academic Editor

PLOS ONE